# Biochemical Intracystic Biomarkers in the Differential Diagnosis of Pancreatic Cystic Lesions

**DOI:** 10.3390/medicina58080994

**Published:** 2022-07-26

**Authors:** Dominika Wietrzykowska-Grishanovich, Ewa Pawlik, Katarzyna Neubauer

**Affiliations:** 1Department of Gastroenterology and Hepatology, University Teaching Hospital, Borowska 213, 50-556 Wroclaw, Poland; ewapawlik93@wp.pl; 2Department of Gastroenterology and Hepatology, Wroclaw Medical University, Borowska 213, 50-556 Wroclaw, Poland

**Keywords:** pancreatic cyst, mucinous cyst, non-mucinous cyst, biomarker, glucose, CEA, pancreatic cancer

## Abstract

*Background and Objectives:* Pancreatic cystic lesions (PCLs) are frequently incidental findings. The prevalence of PCLs is increasing, mainly due to advancements in imaging techniques, but also because of the aging of the population. PCLs comprise challenging clinical problems, as their manifestations vary from benign to malignant lesions. Therefore, the recognition of PCLs is achieved through a complex diagnostic and surveillance process, which in turn is usually long-term, invasive, and expensive. Despite the progress made in the identification of novel biomarkers in the cystic fluid that also support the differentiation of PCLs, their application in clinical practice is limited. *Materials and Methods:* We conducted a systematic review of the literature published in two databases, Pubmed and Embase, on biochemical biomarkers in PCLs that may be applied in the diagnostic algorithms of PCLs. *Results:* Eleven studies on intracystic glucose, twenty studies on intracystic carcinoembryonic antigen (CEA), and eighteen studies on other biomarkers were identified. Low levels of intracystic glucose had high sensitivity and specificity in the differentiation between mucinous and non-mucinous cystic neoplasms. *Conclusions:* CEA and glucose are the most widely studied fluid biochemical markers in pancreatic cystic lesions. Glucose has better diagnostic accuracy than CEA. Other biochemical biomarkers require further research.

## 1. Introduction

Pancreatic cyst lesions (PCLs) are a heterogeneous group of lesions that are characterized by a broad spectrum of behavior, different malignant potential, as well as varying pathologic features. PCLs are frequently incidental findings. The prevalence of PCLs is currently growing, which is not only a reflection of the increased availability and accuracy of abdominal imaging techniques, but also a reflection of the aging population. Nevertheless, PCLs comprise challenging clinical problems, not only because of their increasing occurrence, but mainly because of their heterogeneous manifestations that vary from benign to malignant lesions. At the same time, pancreatic cancer is becoming an progressively common cause of cancer mortality, and a 2.3-fold rise in the global number of cases and deaths from these tumors has been reported [1].

The majority of PCLs are asymptomatic, benign changes that do not require any therapeutic approach. According to their pathological classification, pancreatic cysts are classified into inflammatory fluid collections, non-neoplastic cysts, and pancreatic cystic neoplasms. Inflammatory fluid collections usually result from acute pancreatitis, and are further classified according to the Atlanta criteria into acute peripancreatic fluid collections, pseudocysts, acute necrotic collections, and walled-off pancreatic necrosis. Pancreatic cystic neoplasms (PCNs) are categorized into serous cystic tumors, mucinous cystic neoplasms (MCNs), intraductal papillary mucinous neoplasms (IPMNs), and solid pseudopapillary neoplasms (SPN). PCNs differ in morphology, which is further reflected by their varying appearance in imaging tests, clinical manifestations, and most importantly, their risk of malignant transformation [2] (Figure 1).

Precise recognition of the type of PCL is crucial for the patient’s management. Therefore, the identification of PCLs typically initiates a complex diagnostic and surveillance process, which in turn is usually long-term, invasive, and expensive. In patients with incidentally detected pancreatic cysts, the diagnostic assessment usually includes magnetic resonance imaging (MRI) and magnetic resonance cholangiopancreatography (MRCP), which can best show if the cyst is communicating with the pancreatic duct. For instance, it has been demonstrated that the branched IPMN (BD-IPMN), even though it has malignant potential, may be less malignant compared to the IPMN of the main duct [2]. On the other hand, endoscopic ultrasonography (EUS) can complement diagnostics as the most accurate method of assessing small changes located in the head of the pancreas, enabling the assessment of the cyst structure and its puncture in order to obtain cystic fluid for cytological and biochemical examination [3]. Intracystic fluid biomarkers have an important role in the categorization of PCLs; however, their application in clinical practice is limited to carcinoembryonic antigen (CEA). According to the European evidence-based guidelines on pancreatic cystic neoplasms, assessment of cyst fluid CEA, combined with cytology, or not routinely available KRAS/GNAS mutation analyses, may be considered for differentiating an IPMN or MCN from other PCNs [4].

We conducted a systematic review of biochemical intracystic biomarkers, in order to identify the assortment of fluid indicators that may assist in the diagnostic evaluation of patients with PCLs.

## 2. Materials and Methods

In order to review all studies on the intracystic biomarkers for the differential diagnosis of pancreatic cysts, we searched two publication databases: PubMed and Embase. Combinations of the following keywords were used in queries: “cyst*” AND “pancreatic” AND “pancreas” AND “biomarker*” AND “marker*”. The asterisks allowed us to retrieve records where query words appeared with suffixes (e.g., biomarker|s). The search was limited to publications published between 1 January 2012 and 1 April 2022. No language restrictions were applied, although reports and publications in languages other than English were filtered out in the following curation steps. Duplicate records from the databases were removed before the first eligibility screening was performed. The exclusion criteria were as follows: experimental studies (including animal studies and in vitro research), studies in children, non-original articles, articles on non-biochemical biomarkers (including genetic), studies not on biomarkers, and non-English language articles. Two authors (DW-G and EP) conducted all literature searches. All authors (DW-G, EP, and KN) separately reviewed the titles, the abstracts, as well as the full papers based on the selection criteria, and decided on the suitability of articles for inclusion. All authors then searched the eligible articles. Furthermore, references of the selected papers were cross-searched for omitted relevant articles. Analysis of data was conducted according to PRISMA recommendations. The selection process is presented in Figure 2.

Weighted averages (WA) for two most widely studied markers, CEA and glucose, were calculated using the formula WA = (W1X1 + W2X2 + W3X3 +… + WnXn)/(W1 + W2 + W3 +… + Wn), where w is the number of cases in a single publication and x is the mean for the variable studied.

## 3. Results

Ultimately, 33 articles were included in the systematic review, of which 20 articles assessed intracystic concentrations of carcinoembryonic antigen, and 11 articles assessed intracystic concentrations of glucose. The remaining markers that were included in the systematic review were evaluated among the total number of 19 articles; however, single markers had literature that did not exceed three papers.

Most of the studies were based on groups of patients that did not exceed 100 people. The tested fluid was collected by EUS or during surgical resection of a pancreatic cystic lesion. Only in one study was fluid withdrawn during endoscopic retrograde cholangiopancreatography (ERCP).

### 3.1. Interpretative Synthesis of Data: Carcinoembryonic Antigen

CEA is a non-specific marker whose elevated serum concentration is common in neoplastic diseases, typically in colorectal and pancreatic cancer, and is less common in cancers of the stomach, breast, bronchus, lung, or bladder. However, it may be also present in higher concentrations in non-neoplastic diseases such as hepatitis and liver cirrhosis, chronic pancreatitis, gastric and duodenal ulcer disease, inflammatory bowel diseases, and during pregnancy. CEA cannot be used in cancer screening but using it can help assess the efficacy of oncological treatment, recognize local recurrence as well as distant metastases, and for long-term follow-up of patients after cancer treatment [5].

Among the various markers of cystic fluid (collected during EUS or surgical resection of the lesion), CEA is the most extensively studied. CEA has already found its application in distinguishing mucinous from non-mucinous cysts. Furthermore, it is recommended by the guidelines on pancreatic cystic neoplasms, and is the only biomarker that is widely used in clinical practice [4].

There were 20 papers found that related to intracystic CEA (Table 1). The majority of the identified papers on CEA were retrospective studies (*n* = 13). The numbers of patients included in the studies varied from 17 to 226; however, the majority of papers involved numbers that did not exceed 100 patients. The level of CEA was higher in mucinous cysts than in non-mucinous cysts. The most commonly used cut-off value for the differentiation between mucinous and non-mucinous cysts was 192 ng/mL. The sensitivity at this value ranged from approximately 50 to 75%, with the average being approximately 65%. Few publications took into account other cut-off points; for instance, 317 ng/mL obtained a sensitivity of 89% [6], while for a value of 48 ng/mL, the sensitivity was 72.4% [7]. When referring to the specificity at the cut-off point above 192 ng/mL, it was noted that it ranged from about 80% to 100%, which will most likely distinguish mucinous from non-mucinous lesions. The accuracy ranged from 46% to 84%. WA for CEA was 195 ng/mL.

The combination of CEA with other markers, such as glucose, prostaglandin E-2 (PGE-2), vascular endothelial growth factor A (VEGF-A), and gastricsin, improved the accuracy in distinguishing mucinous from non-mucinous cysts [8,9,10,11,12,13].

As yet, prospective studies involving larger cohorts of patients are necessary, in order to further assess the diagnostic efficacy of such combinations.

### 3.2. Interpretative Synthesis of Data: Glucose

The second most often-studied intracystic biomarker was glucose, which was reported in 11 papers (Table 2). Blood and serum glucose concentration is a simple, available, and broadly used biomarker; it is used routinely in clinical practice. Glucose measurement is quick, easy to perform, and inexpensive. It can be determined by a laboratory test, with a glucometer, or a strip test. It is a repeatable method that requires only a small amount of cyst fluid [5].

The majority of the studies had prospective design (*n* = 6); moreover, the majority of studies (*n* = 6) were published in 2020 and later. The numbers of patients that were included in the evaluated papers ranged from 17 to 153, and the majority of the studies did not exceed 100 patients. The most common cut-off point for glucose was 50 mg/dL. Concentrations below 50 mg/dL were cited for mucinous cysts, and above 50 mg/dL for non-mucinous cysts, with sensitivities, specificities, and accuracies all above 90% (for non-mucinous cysts: 96%, 93.6%, and 94.6%) [24]. WA for glucose was 44 mg/dL. The diagnostic value at the cut-off point of 66 mg/dL had a sensitivity of 94%, a specificity of only 64%, with an accuracy of 84% [16]. The same study also compared glucose to CEA. CEA at the cut-off point above 192 ng/mL had a sensitivity, specificity, and accuracy of 73%, 89%, and 77%, respectively, which demonstrated the advantage of glucose over CEA in terms of sensitivity, specificity, and accuracy in the differentiation between mucinous and non-mucinous cysts.

In nine papers, both glucose and CEA markers were evaluated. Glucose had a significant advantage in sensitivity, specificity, and accuracy over CEA as well as CA 19-9. Thus, glucose may be considered to be used as a routine diagnostic test for pancreatic mucinous cysts. The glucose test is an inexpensive, simple, and broadly available analysis. Whether measured with a laboratory test, a glucometer, or a reagent strip, its levels were found to be significantly lower in mucinous cysts compared to pancreatic non-mucosal cysts. One of the limitations of glucose as a differentiating marker is that pseudocysts, similarly with mucinous cysts, present low glucose levels; only the addition of a second marker, such as CEA, improved diagnostic efficacy [23].

### 3.3. Interpretative Synthesis of Data: Other Biomarkers

Our search identified 19 papers related to biochemical intracystic biomarkers other than CEA and glucose that were studied in pancreatic cystic lesions (Table 3).

Among them was cancer antigen 19-9 (CA 19-9). No common cut-off point that could be applied in practical use was found in the analyzed studies. With the cut-off point above 21.395 kU/L, the specificity, sensitivity, and accuracy were 66%, 78%, and 76%, respectively. CA 19-9 levels were higher in mucinous than in non-mucinous cysts [6]. In the study by Talar-Wojnarowska et al. [15], it was concluded that CA 19-9 levels in pancreatic cyst fluid are less specific compared to CEA, especially in the detection of mucinous cysts. A combination analysis involving several markers simultaneously can improve the accuracy of differential diagnosis. Elevated CA 19-9 levels were found in patients with malignant cysts; low CA 19-9 levels (below 37 U/mL) suggested benign lesions [15].

Another marker that may be useful to analyze pancreatic cyst fluid is amylase. The specificity, sensitivity, and accuracy of amylase were assessed at 80%, 54%, and 68% at the cut-off point of 3.073 U/L. Amylase levels were higher in pseudocysts than in mucinous cysts [6]. In the other study, it was found that the highest level of amylase was associated with pseudocysts, especially in patients with a history of acute pancreatitis [15]. Therefore, use of amylase should be limited to confirm the presence of pseudocysts in patients with a history of pancreatitis.

CA 72-4 is also a marker that deserves attention. In a study that involved a group of over 100 patients, it was found that the CA 72-4 level is higher in mucinous cysts than in non-mucinous cysts, with a high sensitivity of the tests at 94%, quite a low specificity of 73%, and an accuracy of 87% [6]. Similar results were obtained in another study that involved a group of approximately 100 patients [22]. In turn, the combination of CA 72-4 with CEA did not improve the sensitivity, specificity, or the accuracy in distinguishing mucinous from non-mucinous cysts compared to using CEA alone [22].

Another biomarker that can bring us closer to the so-called “perfect marker” is VEGF-A, which had 100% sensitivity, a specificity of over 83%, and an accuracy of close to 100% [12]. It also should be noted that VEGF-A was the only marker that is an exact fluid biomarker for SCN [5]. In addition, the combination of CEA with VEGF-A achieved 95.5% sensitivity and 100% specificity, with 99.3% accuracy. Therefore, VEGF-A requires further intensive research.

Due to knowledge gaps concerning the remaining biomarkers, which result mainly from small numbers of studies on small groups of patients, these markers cannot currently be considered for application in clinical practice. Further research regarding these markers is needed.

## 4. Conclusions

The collection of fluid from the pancreatic cyst in EUS and further biochemical analysis of its composition are very useful in assessing the risk of malignancy, and for making further diagnostic and therapeutic decisions. Despite the progress made in the identification of novel biomarkers in the cystic fluid and supporting the differentiation of the PCLs, their application in clinical practice is still limited to CEA. CEA is the best-tested biomarker thus far; however, it is glucose that has greater sensitivity, specificity, and accuracy in distinguishing mucinous from non-mucinous cysts. Using a combination of the two markers is the most effective. Unfortunately, other biomarkers identified in this systematic review require further research, as data on their diagnostic potential are still limited and they cannot be in widespread use. Therefore, it should be highlighted that the glucose assay, which is available, simple, and inexpensive, could serve as a differentiating marker in clinical practice, despite the mechanisms leading to different glucose levels in different cyst types being unclear. Glucose, due to its diagnostic efficiency and simple measurement, may in the future replace CEA in the differential diagnosis of cysts. However, among the limitations of the existing studies on glucose in PCLs, the different cut-off values applied by the authors must be considered. Most probably, only a combination of several markers may bring us closer to making correct diagnoses. For instance, simultaneous determination of glucose and CEA levels resulted in higher sensitivity.

In summary, the perfect biochemical marker that allows for precise classification and risk stratification of pancreatic cystic lesions is still not available, and the existing indices play a supplementary role in the diagnostic process (Figure 3). Considering the growing number of patients with PCLs and their consequences, which involve multiple imaging tests and surgical interventions, further research is needed to find a novel single marker or a panel of markers that would allow health care personnel to clearly define the nature of the examined lesion and exclude its malignancy. This may help save patients from unnecessary surgeries, while qualifying others for appropriate treatment at an early stage of the disease.

## Figures and Tables

**Figure 1 medicina-58-00994-f001:**
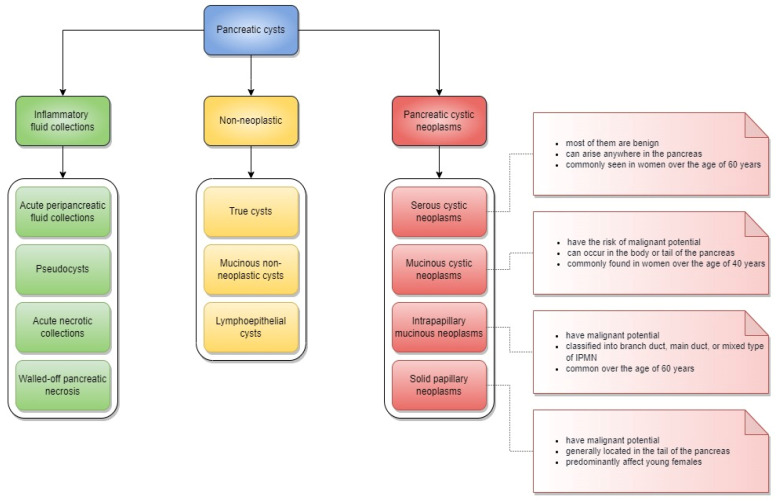
Pathological classification of pancreatic cysts and clinical features of pancreatic cystic neoplasms. IPMN, intraductal papillary mucinous neoplasms.

**Figure 2 medicina-58-00994-f002:**
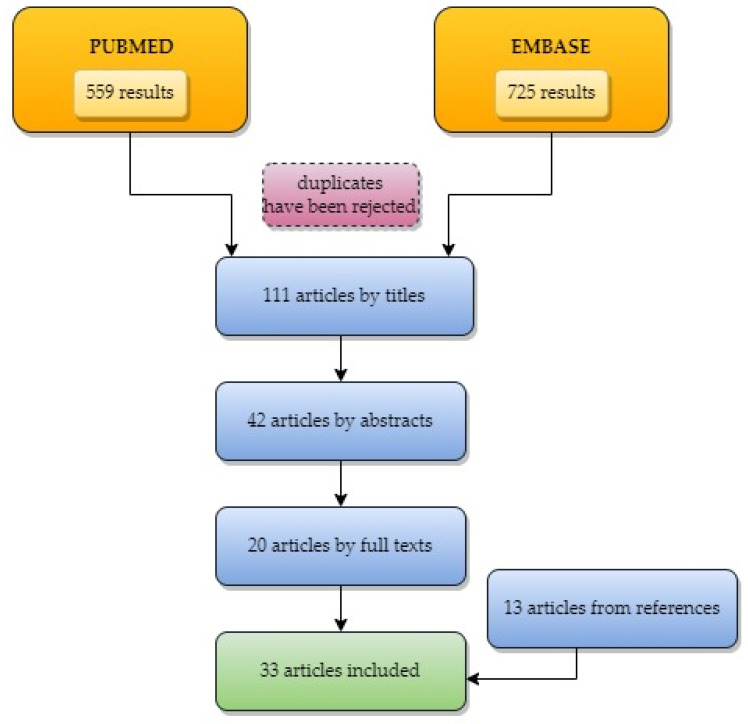
Flowchart presenting the selection process.

**Figure 3 medicina-58-00994-f003:**
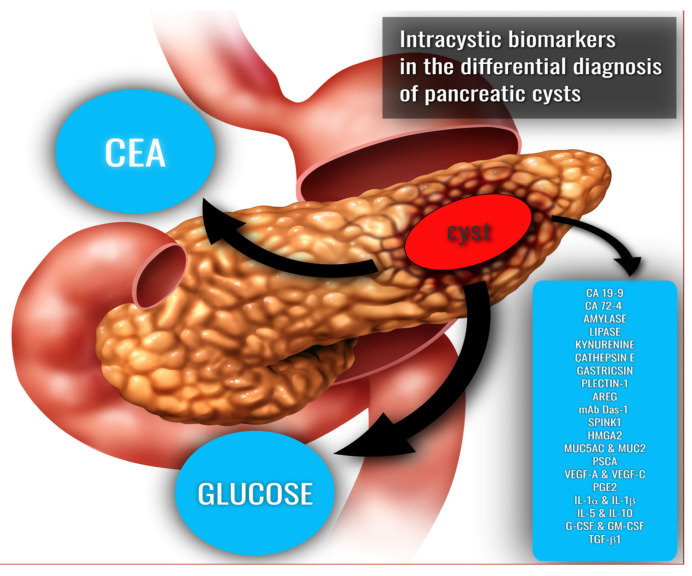
Biochemical intracystic biomarkers in pancreatic cystic lesions. CEA, carcinoembryonic antigen; CA 19-9, cancer antigen 19-9; CA 72-4, cancer antigen 72-4; AREG, amphiregulin; mAb Das-1, monoclonal antibody against a colonic epithelial antigen; SPINK1, serine protease inhibitor Kazal-type 1; HMGA2, high-mobility group AT-hook 2; MUC5AC, mucin 5AC; MUC2, mucin 2; PSCA, prostate stem cell antigen; VEGF-A, vascular endothelial growth factor-A; VEGF-C, vascular endothelial growth factor-C; PGE2, prostaglandin E2; IL-1α, interleukin 1 alpha; IL-1β, interleukin 1 beta; IL-5, interleukin 5; IL-10, interleukin 10; G-CSF, granulocyte colony-stimulating factor; GM-CSF, granulocyte-macrophage colony-stimulating factor; TGF-β1, transforming growth factor β1. An image made by Lightspring/Shutterstock.com was used to create this graphic.

**Table 1 medicina-58-00994-t001:** Intracystic CEA levels in the differentiation of pancreatic cystic lesions.

Author	Year	Study	Patients (*n*)	Sensitivity (%)	Specificity (%)	Accuracy (%)	Main Findings
Kucera S. et al. [14]	2012	retrospective cross-sectional study EUS-FNA	47	CEA > 200 ng/mL 52.4% for IPMN	CEA > 200 ng/mL 42.3% for IPMN	CEA > 200 ng/mL 46.8% for IPMN	The mean levels of CEA increased as pathology progressed from low-grade dysplasia to moderate and high-grade dysplasia. The mean CEA level decreased when invasive cancer developed.
Talar - Wojnarowska R. et al. [15]	2012	prospective study EUS-FNA	52	CEA cut-off point 45 ng/mL 91.8%	CEA cut-off point 45 ng/mL 63.9%	CEA cut-off point 45 ng/mL 89.2%	CEA was higher in patients with malignant cysts compared to benign lesions.
Park W.G. et al. [16]	2013	retrospective cohort study EUS-FNA and surgical resection	31 from 45	CEA > 192 ng/mL 73%	CEA > 192 ng/mL 89%	CEA > 192 ng/mL 77%	CEA > 192 ng/mL in combination with glucose < 66 mg/dl showed better diagnostic accuracy in differentiating mucinous from non-mucinous cysts compared to the above markers alone.
Nagashio Y. et al. [17]	2014	retrospective study EUS-FNA and surgical resection	68	CEA cut-off point 67.3 ng/mL 89.2%	CEA cut-off point 67.3 ng/mL 77.8%	CEA cut-off point 67.3 ng/mL 88.4%	CEA can be a helpful marker in differentiating mucinous from non-mucinous cysts, but not malignant from benign cystic lesions.
Yadav D. et al. [8]	2014	retrospective study fluid aspiration method was not mentioned	17	CEA ≥ 184 ng/mL 36% CEA ≥ 184 ng/mL with glucose ≤ 21 mg/dL 100%	CEA ≥ 184 ng/mL 100% CEA ≥ 184 ng/mL with glucose ≤ 21 mg/dL 83%	CEA ≥ 184 ng/mL 70% no data	Patients with non-mucinous cysts (pseudocysts) had higher levels of intracystic glucose. The differentiation based on CEA levels was not that good. The use of a combination of glucose ≤ 21 or CEA ≥ 184 did not improve diagnoses.
Gaddam S. et al. [18]	2015	retrospective study surgical resection	226	CEA cut-off point 105 ng/mL 70% CEA cut-off point 192 ng/mL 61%	CEA cut-off point 105 ng/mL 63% CEA cut-off point 192 ng/mL 77%	CEA cut-off point 105 ng/mL 77% CEA cut-off point 192 ng/mL 61%	CEA had clinically suboptimal accuracy in distinguishing MCN from NMCN.
Jin D.X. et al. [19]	2015	retrospective study surgical resection	86	no data	no data	CEA cut-off point 30.7 ng/mL 87.2% for differentiating mucinous from non-mucinous cysts CEA cut-off point 30.7 ng/mL 82.7% for differentiating IPMN from non-mucinous cysts	CEA level was significantly higher in mucinous cysts compared with non-mucinous cysts and in IPMN compared with non-mucinous cysts. CEA levels were not significantly different between malignant and non-malignant mucinous cysts.
Zikos T. et al. [9]	2015	prospective study methods of collecting fluid from the cysts were not mentioned	65	CEA > 192 ng/mL 77% CEA > 192 ng/mL with glucose < 50 mg/dL 100%	CEA > 192 ng/mL 83% CEA > 192 ng/mL with glucose < 50 mg/dL 33%	no data	CEA in combination with glucose showed greater sensitivity but less specificity than using CEA alone. Glucose, whether measured with a laboratory test, glucometer, or reagent strip, was significantly lower in mucinous cysts compared to non-mucosal cysts.
Oh S.H. et al. [7]	2016	retrospective study EUS-FNA	48	CEA cut-off point 48.6 ng/mL 72.4%	CEA cut-off point 48.6 ng/mL 94.7%	CEA cut-off point 48.6 ng/mL 81.3%	CEA was the best single test for identifying mucinous cysts. The addition of cytology and string symptom assessment to the fluid CEA increased the overall accuracy in the diagnosis of mucinous cysts.
Carr R.A. et al. [12]	2017	retrospective study EUS-FNA and surgical resection and ERCP	149	CEA ≤ 10 ng/mL 95.5%	CEA ≤ 10 ng/mL 81.5%	CEA ≤ 10 ng/mL 94.5%	VEGF-A was a very accurate test for SCN. The combination of VEGF-A and CEA approached the gold standard in the diagnosis of pancreatic lesions.
Carr R.A. et al. [20]	2017	retrospective study EUS-FNA and surgical resection	153	CEA > 192 ng/mL 58%	CEA > 192 ng/mL 96%	CEA > 192 ng/mL 69%	Glucose had a significant diagnostic advantage over CEA.
Ivry S.L. et al. [13]	2017	retrospective study EUS-FNA and surgical resection	89	CEA cut-off point 192 ng/mL 65%	CEA cut-off point 192 ng/mL 94%	CEA cut-off point 192 ng/mL 86.5%	CEA was significantly elevated in the mucinous cysts. The activities of cathepsin E and gastricsin strongly increased in the fluid of mucinous vs. non-mucinous cysts. Best results were achieved when gastricsin and CEA were combined.
Jabbar K.S. et al. [21]	2017	prospective cohort study EUS-FNA	105	CEA cut-off point 1000 ng/mL 54%	CEA cut-off point 1000 ng/mL 90%	CEA cut-off point 1000 ng/mL 84%	MUC5AC plus PSCA yielded a significantly higher percentage of correct HGD/cancer scores than CEA and cytology.
Levy A. et al. [6]	2017	retrospective study EUS-FNA	115	CEA cut-off point 317 µg/L 89%	CEA cut-off point 317 µg/L 93%	CEA cut-off point 317 µg/L 93%	CEA in cyst fluid was higher in mucinous cysts than in non-mucinous ones.
Soyer O.M. et al. [22]	2017	retrospective cohort study EUS-FNA	96	CEA cut-off point 207 ng/mL 72.7%	CEA cut-off point 207 ng/mL 97.7%	CEA cut-off point 207 ng/mL 89.5%	CEA and CA 72.4 levels for benign-mucinous and malignant cysts were significantly higher than for non-mucinous cysts. The levels of CEA and CA 72-4 in the cystic fluid are highly accurate in distinguishing mucinous from non-mucinous cysts, but with cytology, their accuracy increases.
Faias S. et al. [23]	2019	retrospective study EUS-FNA	82	CEA > 192 ng/mL 72%	CEA > 192 ng/mL 96%	CEA > 192 ng/mL 84.2%	Pseudocysts presented low glucose identically to mucinous cysts; only glucose with CEA allowed differential diagnosis.
Ribaldone D.G. et al. [24]	2020	prospective study EUS-FNA	56	CEA > 192 ng/mL 54.8% for mucinous cysts CEA < 5 ng/mL 72% for non-mucinous cysts	CEA > 192 ng/mL 100% for mucinous cysts CEA < 5 ng/mL 87.1% for non-mucinous cysts	CEA > 192 ng/mL 75% for mucinous cysts CEA < 5 ng/mL 80.4% for non-mucinous cysts	Glucose was more sensitive than CEA in the differential diagnosis of mucinous versus non-mucinous pancreatic cysts.
Rossi G. et al. [25]	2020	prospective study EUS-FNA	48	CEA ≥ 192 ng/mL 37.5%	CEA ≥ 192 ng/mL 100%	CEA ≥ 192 ng/mL 69%	Glucose was a valid and simple tool for the differential diagnosis of mucinous vs. non-mucinous lesions. It was more accurate than CEA levels.
Simons-Linares C.R. et al. [10]	2020	prospective cohort study EUS-FNA	113	CEA ≥ 192 ng/mL 50% CEA ≥ 192 ng/mL with glucose ≤ 21 mg/dl 93%	CEA ≥ 192 ng/mL 92% CEA ≥ 192 ng/mL with glucose ≤ 21 mg/dL 92%	no data	Glucose outperformed CEA for differentiating mucinous from non-mucinous pancreatic cysts.
Smith Z. L. et al. [26]	2022	prospective cohort study EUS-FNA	93	CEA ≥ 192 ng/mL 62.7%	CEA ≥ 192 ng/mL 88.2%	CEA ≥ 192 ng/mL 81%	Glucose was superior to CEA for differentiating MCNP when analyzed from freshly obtained fluid of cysts with histologic diagnoses.

CEA, carcinoembryonic antigen; EUS-FNA, endoscopic ultrasound-guided fine-needle aspiration; IPMN, intraductal papillary mucinous neoplasm; MCN, mucinous cystic neoplasm; NMCN, non-mucinous cystic neoplasm; CA 19-9, cancer antigen 19-9; VEGF-A, vascular endothelial growth factor-A; SCN, serous cystic neoplasm; MUC5AC, mucin 5AC; PSCA, prostate stem cell antigen; HGD, high-grade dysplasia; CA 72-4, cancer antigen 72-4; MCNP, mucinous cystic neoplasms of the pancreas.

**Table 2 medicina-58-00994-t002:** Intracystic glucose levels in differentiating pancreatic cystic lesions.

Author	Year	Study	Patients (n)	Sensitivity (%)	Specificity (%)	Accuracy (%)	Main Findings
Park W.G. et al. [16]	2013	retrospective cohort study EUS-FNA and surgical resection	26—I cohort 19—II cohort together 45	glucose cut-off point 66 mg/dL 94%	glucose cut-off point 66 mg/dL 64%	glucose cut-off point 66 mg/dL 88%	Metabolomic abundance for glucose and kynurenine was significantly lower in mucinous cysts compared to non-mucinous cysts. Neither could differentiate premalignant from malignant cysts. Glucose and kynurenine levels were significantly elevated for serous cystadenomas.
Yadav D. et al. [8]	2014	retrospective study fluid aspiration method was not mentioned	17	glucose ≤ 21 mg/dL 100%	glucose ≤ 21 mg/dL 83%	glucose ≤ 21 mg/dL 87%	Patients with non-mucinous cysts (pseudocysts) had higher levels of intracystic glucose.
Zikos T. et al. [9]	2015	prospective study methods of collecting fluid from the cysts were not mentioned	65	laboratory—glucose < 50 mg/dL 95% glucometer—glucose < 50 mg/dL 88% reagent strip — glucose 81%	laboratory—glucose < 50 mg/dL 57% glucometer—glucose < 50 mg/dL 78% reagent strip — glucose 74%	no data	Glucose, whether measured with a laboratory test, glucometer, or reagent strip, was significantly lower in mucinous cysts compared to pancreatic non-mucosal cysts.
Carr R.A. et al. [20]	2017	retrospective study EUS-FNA and surgical resection	153	glucose ≤ 50 mg/dL 92% for mucinous cysts	glucose ≤ 50 mg/dL 87% for mucinous cysts	glucose ≤ 50 mg/dL 90% for mucinous cysts	Glucose in the cystic fluid was lower in mucinous cysts compared to non-mucinous cysts. Glucose outperformed CEA.
Faias S. et al. [23]	2019	retrospective study EUS-FNA	82	glucose < 50 mg/dL 89%	glucose < 50 mg/dL 86%	glucose < 50 mg/dL 86%	Pseudocysts presented low glucose, identically to mucinous cysts. Glucose combined with CEA allowed differential diagnosis.
Oria I. et al. [27]	2020	prospective study EUS-FNA	75	glucose ≤ 50 mg/dL 89.4%	glucose ≤ 50 mg/dL 76.2%	glucose ≤ 50 mg/dL 84%	Glucose was a very accurate, rapid, and inexpensive test for the diagnosis of mucinous PCLs.
Ribaldone D.G. et al. [24]	2020	prospective study EUS-FNA	56	glucose < 50 mg/dL 93.6% for mucinous cysts glucose ≥ 50 mg/mL 96% for non-mucinous cysts	glucose < 50 mg/dL 96% for mucinous cysts glucose ≥ 50 mg/mL 93.6% for non-mucinous cysts	glucose < 50 mg/dL 94.6% for mucinous cysts glucose ≥ 50 mg/mL 94.6% for non-mucinous cysts	Glucose was more sensitive than CEA in the differential diagnosis of mucinous versus non-mucinous pancreatic cysts.
Rossi G. et al. [25]	2020	prospective study EUS-FNA	48	glucose ≤ 30 mg/dL 91.3%	glucose ≤ 30 mg/dL 100%	glucose ≤ 30 mg/dL 95%	Glucose level in the cyst fluid obtained during EUS with FNA represented a valid and simple tool for the differential diagnosis of mucinous vs. non-mucinous lesions and was more accurate than CEA.
Simons-Linares C.R. et al. [10]	2020	prospective cohort study EUS-FNA	113	glucose ≤ 41 mg/dL 92% glucose ≤ 21 mg/dL 88%	glucose ≤ 41 mg/dL 92% glucose ≤ 21 mg/dL 97%	glucose ≤ 41 mg/dL 95% no data	Glucose outperformed CEA for differentiating mucinous from non-mucinous pancreatic cysts.
Noia J. L. et al. [28]	2021	retrospective study EUS-FNA	72 (40 in the derivation cohort and 32 in the validation cohort)	glucose cut-off point 73 mg/dL 89% for derivation cohort 100% for validation cohort	glucose cut-off point 73 mg/dL 90% for derivation cohort 71% for validation cohort	no data	On-site glucometry was a feasible, accurate, and reproducible method for the characterization of PCLs after EUS-FNA. It showed an excellent correlation with laboratory glucose values.
Smith Z. L. et. al. [26]	2022	prospective cohort study EUS-FNA	93	glucose ≤ 25 mg/dL 88.1%	glucose ≤ 25 mg/dL 91.2%	glucose ≤ 25 mg/dL 96%	Glucose was superior to CEA for differentiating MCNP when analyzed from the freshly obtained fluid of cysts with histologic diagnoses.

EUS-FNA, endoscopic ultrasound-guided fine-needle aspiration; CA 19-9, cancer antigen 19-9; CEA, carcinoembryonic antigen; PCLs, pancreatic cystic lesions; MCNP, mucinous cystic neoplasms of the pancreas.

**Table 3 medicina-58-00994-t003:** Other intracystic biomarkers used in differentiating pancreatic cystic lesions.

Author	Year	Marker	Study	Patients (*n*)	Sensitivity (%)	Specificity (%)	Accuracy (%)	Main Findings
Lee L.S. et al. [29]	2012	TGF-β1 G-CSF	prospective study EUS-FNA and ERCP	10	no data	no data	no data	Intracystic TGF-β1 and G-CSF were suggested to be potential diagnostic biomarkers that could distinguish mixed IPMN from BD-IPMN.
Talar - Wojnarowska R. et al. [15]	2012	CA 19-9	prospective study EUS-FNA	52	CA 19-9 cut-off point 37 U/mL 81.3%	CA 19-9 cut-off point 37 U/mL 69.4%	CA 19-9 cut-off point 37 U/mL 87.3%	CA 19-9 was considered to be less specific compared to CEA, particularly for the detection of mucinous cysts. CA 19-9 had higher sensitivity and specificity than CEA in the detection of pancreatic cystadenocarcinomas.
Talar - Wojnarowska R. et al. [15]	2012	amylase	prospective study EUS-FNA	52	amylase 62.5%	amylase 69.4%	amylase 68.4%	Amylase can be useful for the confirmation of pseudocyst diagnosis, particularly in patients with a history of pancreatitis. Mean amylase levels in benign lesions were higher compared to malignant cysts.
Tun M.T. et al. [30]	2012	AREG	retrospective study EUS-FNA surgical resection	33	AREG > 300 pg/mL 83% for cancer or high-grade dysplasia	AREG > 300 pg/mL 73% for cancer or high-grade dysplasia	AREG > 300 pg/mL 78% for cancer or high-grade dysplasia	AREG levels were significantly higher in cancerous and high-grade dysplastic cysts compared to benign mucinous cysts.
Das K.K. et al. [31]	2013	mAb Das-1	retrospective cohort study EUS-FNA and surgical resection	94 + 38	mAb Das-1 in high risk/malignant IPMNs 85% in resection tissue 89% in liquid from EUS-FNA	mAb Das-1 in high risk/malignant IPMNs 95% in resection tissue 100% in liquid from EUS-FNA	no data	mAb Das-1 reacted with high specificity to tissue and cyst fluid from high-risk/malignant IPMNs.
Park W.G. et al. [16]	2013	kynurenine	retrospective cohort study EUS-FNA and surgical resection	26—I cohort 19—II cohort	kynurenine cut-off point 185,650 100% kynurenine cut-off point 34,000 90%	kynurenine cut-off point 185,650 80% kynurenine cut-off point 34,000 100%	kynurenine cut-off point 185,650 94% kynurenine cut-off point 34,000 92%	Kynurenine levels were significantly elevated in SCA lesions compared to lesions that were not SCAs.
Räty S. et al. [32]	2013	SPINK1	prospective study surgical resection	61	SPINK1 cut-off point 118 μg/L 85% for differentiating MCA or main/mixed type IPMN from SCA or side branch IPMN SPINK1 cut-off point 146 μg/L 93% for differentiating < 3 cm MCA or main duct IPMN from SCA or side branch IPMN	SPINK1 cut-off point 118 μg/L 84% for differentiating MCA or main/mixed type IPMN from SCA or side branch IPMN SPINK1 cut-off point 146 μg/L 89% for differentiating < 3 cm MCA or main duct IPMN from SCA or side branch IPMN	SPINK1 cut-off point 118 μg/L 94% for differentiating MCA or main/mixed type IPMN from SCA or side branch IPMN SPINK1 cut-off point 146 μg/L 98% for differentiating < 3 cm MCA or main duct IPMN from SCA or side branch IPMN	SPINK1 may be a possible marker in the differential diagnosis of benign and potentially malignant pancreatic cystic lesions.
Yip-Schneider M.T. et al. [33]	2014	VEGF-A VEGF-C	prospective study surgical resection	87	VEGF-A cut-off point 8500 pg/mL 100% VEGF-C cut-off point 200 pg/mL 100%	VEGF-A cut-off point 8500 pg/mL 97% VEGF-C cut-off point 200 pg/mL 90%	no data	VEGF-A and VEGF-C were significantly upregulated in SCN compared with all other diagnoses.
DiMaio C.J. et al. [34]	2015	HMGA2 protein	retrospective study surgical resection	31	no data	no data	no data	Significantly higher concentrations of HMGA2 protein in the cystic fluid were found in IPMN with HGD compared to changes with LGD or MD.
Moris M. et al. [35]	2016	plectin-1	retrospective study EUS-FNA and surgical resection	104	plectin-1 75% in PDA vs. non- PDA IPMNs	plectin-1 85% in PDA vs. non- PDA IPMNs	plectin-1 79% in PDA vs. non- PDA IPMNs	Plectin-1 distinguished IPMN with invasive adenocarcinoma from non-invasive IPMN, but was insufficient for discriminating HGD IPMN from LGD IPMNs.
Carr R.A. et al. [12]	2017	VEGF-A	retrospective study EUS-FNA, surgical resection, ERCP	149	VEGF-A > 5000 pg/mL 100% VEGF-A with CEA 99.5%	VEGF-A > 5000 pg/mL 83.7% VEGF-A with CEA 100%	VEGF-A > 5000 pg/mL 98.3% VEGF-A with CEA 99.3%	Although VEGF-A was a very accurate test for SCN, a combination of VEGF-A and CEA approached the gold standard in the diagnosis of pancreatic lesions.
Ivry S.L. et al. [13]	2017	sathepsin E gastricsin	retrospective study EUS-FNA and surgical resection	110	cathepsin E 70% gastricsin 93% gastricsin with CEA 98%	cathepsin E 92% gastricsin 100% gastricsin with CEA 100%	cathepsin E 82.8% gastricsin 97.9% gastricsin with CEA 99.8%	Activity of cathepsin E and gastricsin increased in the fluid of mucinous vs. non-mucinous cysts; the best results were obtained when combined with gastricsin and CEA.
Jabbar K.S. et al. [21]	2017	MUC5AC with PSCA	Prospective cohort study EUS-FNA	105 68	MUC5AC with PSCA cut-off point 12 fmol/µL (summed protein concentration levels) 95% MUC5AC with MUC2 cut-off point 0.01 fmol/µL (summed protein concentration levels) 96%	MUC5AC with PSCA cut-off point 12 fmol/µL (summed protein concentration levels) 96% MUC5AC with MUC2 cut-off point 0.01 fmol/µL (summed protein concentration levels) 100%	MUC5AC with PSCA cut-off point 12 fmol/µL (summed protein concentration levels) 96% MUC5AC with MUC2 cut-off point 0.01 fmol/µL (summed protein concentration levels) 97%	MUC5AC plus PSCA achieved a significantly higher percentage of correct HGD/cancer scores than CEA and cytology. Panel of peptides from mucin-5AC and mucin-2 could discriminate premalignant/malignant lesions from benign.
Levy A. et al. [6]	2017	CA 19-9	retrospective study EUS-FNA	115	CA 19-9 cut-off point 21.395 kU/l 66%	CA 19-9 cut-off point 21.395 kU/L 78%	CA 19-9 cut-off point 21.395 kU/L 76%	CA 19-9 was higher in mucinous cysts than non-mucinous ones.
Levy A. et al. [6]	2017	CA 72-4	retrospective study EUS-FNA	115	CA 72-4 cut-off point 7.0 kU/l 94%	CA 72-4 cut-off point 7.0 kU/L 73%	CA 72-4 cut-off point 7.0 kU/L 87%	CA 72-4 was higher in mucinous cysts than in non-mucinous cysts.
Levy A. et al. [6]	2017	amylase	retrospective study EUS-FNA	115	amylase cut-off point 3.073 U/L 80%	amylase cut-off point 3.073 U/L 54%	amylase cut-off point 3.073 U/L 68%	Amylase levels, which indicate pancreatic duct communication, were higher in PCs than in mucinous cysts.
Levy A. et al. [6]	2017	lipase	retrospective study EUS-FNA	115	lipase cut-off point 39.260 U/L 88%	lipase cut-off point 39.260 U/l 45 %	lipase cut-off point 39.260 U/L 63%	Lipase levels, which indicate pancreatic duct communication, were higher in PCs than in mucinous cysts.
Soyer O.M. et al. [22]	2017	CEA and CA 72-4	retrospective cohort study EUS-FNA	96	CA 72-4 cut-off point 3.32 ng/mL 80%	CA 72-4 cut-off point 3.32 ng/mL 69.5%	CA 72-4 cut-off point 3.32 ng/mL 73.6%	CEA and CA 72-4 levels for benign-mucinous and malignant cysts were significantly higher than for non-mucinous cysts. Levels of CEA and CA 72-4 in the cystic fluid were highly accurate in distinguishing mucinous from non-mucinous cysts; with cytology, their accuracy increases further.
Yip-Schneider M.T. et al. [11]	2017	PGE-2	prospective study surgical resection	100	PGE2 cut-off point 1.1 pg/µL 63% PGE2 cut-off point 0.5 pg/µL with CEA > 192 ng/mL 78%	PGE2 cut-off point 1.1 pg/µL 79% PGE2 cut-off point 0.5 pg/µL with CEA > 192 ng/mL 100%	PGE2 cut-off point 1.1 pg/µL 71% PGE2 cut-off point 0.5 pg/µL with CEA > 192 ng/mL 86%	PGE2 was an indicator of IPMN dysplasia, especially in selected patients with preoperative pancreatic cyst fluid CEA > 192ng/mL. PGE2 levels in high-grade/invasive IPMN were significantly higher than in low/moderate-grade IPMN.
Das K.K. et al. [36]	2019	mAb Das-1	retrospective study surgical resection	169	mAb Das-1 cut-off optical density value 0.104 88%	mAb Das-1 cut-off optical density value 0.104 99%	mAb Das-1 cut-off optical density value 0.104 95%	Authors validated the ability of an ELISA with the monoclonal antibody Das-1 to detect PCLs at risk for malignancy with high levels of sensitivity and specificity.
Simpson R.E. et al. [37]	2019	IL-1β and PGE2	retrospective study EUS-FNA and surgical resection	92	IL-1 β > 20 pg/mL 64.3% PGE2 > 1100 pg/mL 60% IL-1β with PGE2 42.9%	IL-1 β > 20 pg/mL 83.8% PGE2 > 1100 pg/mL 78.7% IL-1β with PGE2 89.2%	IL-1 β > 20 pg/mL 73.4% PGE2 > 1100 pg/mL 69.6% IL-1β with PGE2 64.6%	IL-1β and PGE2 levels were higher in high-grade/invasive IPMN than in low/moderate-grade IPMN.
Siu L. et al. [38]	2019	IL-1α, IL-5, IL-10, and GM-CSF	prospective study EUS-FNA and surgical resection	23	no data	no data	no data	IL-1α and IL-5 had higher concentrations in non-mucinous cysts, while IL-10 and GM-CSF had higher concentrations in mucinous cysts.

EUS-FNA, endoscopic ultrasound-guided fine-needle aspiration; ERCP, endoscopic retrograde cholangiopancreatography; TGF-β1, transforming growth factor β1; G-CSF, granulocyte colony-stimulating factor; IPMN, intraductal papillary mucinous neoplasm; BD-IPMN, branch duct-intraductal papillary mucinous neoplasm; CA 19-9, cancer antigen 19-9; CEA, carcinoembryonic antigen; AREG, amphiregulin; mAb Das-1, monoclonal antibody against a colonic epithelial antigen; SCA, serous cystadenoma; SPINK1, serine protease inhibitor Kazal-type 1; MCA, mucinous cystadenoma; VEGF-A, vascular endothelial growth factor-A; VEGF-C, vascular endothelial growth factor-C; SCN, serous cystic neoplasm; HMGA2, high-mobility group AT-hook 2; HGD, high-grade dysplasia; LGD, low-grade dysplasia; MD, main-duct; PDA, pancreatic ductal adenocarcinoma; non-PDA, non-pancreatic ductal adenocarcinoma; MUC5AC, mucin 5AC; MUC2, mucin 2; PSCA, prostate stem cell antigen; CA 72-4, cancer antigen 72-4; PC, pancreatic cancer; PGE2, prostaglandin E2; ELISA, enzyme-linked immunosorbent assay; PCL, pancreatic cystic lesion; IL-1β, interleukin 1 beta; IL-1α, interleukin 1 alpha; IL-5, interleukin 5; IL-10, interleukin 10; GM-CSF, granulocyte-macrophage colony-stimulating factor.

## Data Availability

Not applicable.

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
