# Peer review of "Biochemical Intracystic Biomarkers in the Differential Diagnosis of Pancreatic Cystic Lesions"

_medicina, 2022, doi:10.3390/medicina58080994_

Round 1

Reviewer 1 Report

I have the following minor remarks:

a.       abstract should have subsections (background, material and methods, results and conclusions),

b.       if the authors performed the review according to the PRISMA guidelines it should be stated,

c.       in section conclusions -  the supplementary role of markers should be highlighted as they assist in the diagnosis-making process, and the final diagnosis does not rely solely on them,

d.       in section conclusions – the limitations of the available studies should be summarised.

Reviewer 2 Report

The manuscript is very interesting by the topic, design and the volume of analyzed information. 

It is useful for understanding of the problem of diagnostic of pancreatic cystic lesions.

The article provides data on the laboratory assessment of the content of cystic lesions of the pancreas, which is an important and not fully understood issue in the differential diagnosis of pancreatic cysts. The authors did a great job of analyzing and synthesizing the results of the studies published to date and presented a systematic review. The authors reasonably showed that the most informative markers of mucinous cystadenomas, as the most threatened pancreatic cysts in relation to malignancy, are carcinoembryonic antigen and glucose. The remaining markers (CA 19-9B, CA 72-4, amylase and other molecules) demonstrate significantly lower sensitivity and specificity.

There are no significant comments on the article, but there are a few small remarks.

In the section on glucose, the authors indicate cut-off values, sensitivity, specificity, and accuracy without mentioning which variant of cystic formation they are talking about.

The authors discuss only two options as relevant markers, while citing quite a lot of studies confirming this judgment. At the same time, the cut-off value of these markers differs between studies, which makes it difficult to understand which data should be relied upon in the first place. Therefore, as a suggestion for improving the article, we invite the authors to consider the method of calculating weighted averages, which was used by a number of authors when writing systematic reviews. Formula for the calculation and the source are given below.

“A weighted average (WA) is used to obtain a statistical sum of all the means for the different variables….” 

The following formula was used

WA = (W1X1 + W2X2 + W3X3 +… + WnXn)/(W1 + W2 + W3 +… + Wn) ,

Gumbs AA, Rodriguez Rivera AM, Milone L, Hoffman JP (2011) Laparoscopic pancreatoduodenectomy: a review of 285 published cases. Ann Surg Oncol 18:1335–1341

Сalculation of weighted averages will allow to determine the cut-off values of carcinoembryonic antigen and glucose, taking into account all published data

 Otherwise, the article deserves a good rating and needs to be published after minor corrections.
